

# Intra-session reliability of isometric muscle strength of the bilateral standing press in female handball players

Claudio Cifuentes-Zapata[1,2], Oscar Andrades-Ramírez[3], David Ulloa-Díaz[4], Ángela Rodríguez-Perea[1], Álvaro Huerta Ojeda[5] and Luis Javier Chirosa-Ríos[1]

[1] Department of Physical Education and Sport, Faculty of Sport Sciences, University of Granada, Granada, Andalucía, Spain
[2] Núcleo de Investigación en Salud, Actividad Física y Deportes ISAFYD, Escuela de Ciencias de la Actividad Física, Universidad de las Américas, Santiago, Metropolitana, Chile
[3] Facultad de Educación y Ciencias Sociales, Universidad Andres Bello, Concepción, Bio Bio, Chile
[4] Department of Sports Sciences and Physical Conditioning, Universidad Católica de la Santísima Concepción, Concepción, Bio Bio, Chile
[5] Facultad de Educación, Núcleo de Investigación en Salud, Actividad Física y Deporte ISAFYD, Viña del Mar, Valparaiso, Chile

Corresponding author
David Ulloa-Díaz, dulloa@ucsc.cl

## ABSTRACT

**Background:** Systematizing reliable protocols and procedures for strength assessment in handball has allowed for a more thorough kinetic analysis and increased precision in detecting training-induced changes in muscular strength. The evaluation of upper limb strength with a bilateral standing press (BSP) exercise in handball players approximates blocking actions in the defensive phase, pushing and fixation.

**Aim:** The aim of this study was to analyze the relative and absolute reliability of intra-session comparisons in an isometric peak muscle strength protocol in a bilateral standing press (BSP) exercise among female handball players.

**Methods:** Sixteen young female handball players at an international level, aged between 22 ± 4 years, with no prior experience in using functional electromechanical dynamometers (FEMD) participated in this study. The participants initiated the assessment of maximum isometric force in a bipedal stance with a forward projection of the dominant foot between 20 and 30 cm. The knees were kept semi-flexed, and the hip extension of the non-dominant limb ranged from 15 to 20°. Bilateral anterior push of the upper limb was performed with shoulder abduction and elbow flexion at 90°, maintaining a pronated hand position. Participants were instructed to exert three sets of maximal force for 5 s in BSP exercise. Relative reliability was assessed using the model intraclass correlation (ICC) and absolute reliability was assessed using the coefficient of variation (CV) and standard error of measurement (SEM). For this study, the parameters of maximum and mean muscle strength were considered.

**Results:** The results demonstrated high relative reliability (ICC 0.93–0.97) and absolute reliability (SEM 0.19–2.79) y (CV 4.78–9.03) for both mean force and peak force, with no significant differences between the sets ($p > 0.05$), indicating a negligible effect size (0.01–0.12).
**Conclusion:** The mean and peak isometric muscle strength for the BSP exercise controlled with FEMD in female handball players exhibits high relative and absolute reliability between series.

# INTRODUCTION

Systematizing reliable protocols and procedures for evaluating various manifestations of muscular strength has been of great interest in the fields of medicine and sports performance (*Janicijevic et al., 2023*; *Ortega-Becerra et al., 2017*; *Weakley et al., 2021*). This has facilitated a more comprehensive kinetic analysis and increased precision in detecting training-induced changes, thereby enhancing the role of coaches and strength and conditioning professionals in providing valuable feedback for the training and competition process.

Most studies indicate that muscular strength is a crucial determinant of sports performance, especially in physically demanding sports like handball, requiring high levels of speed and muscular strength for optimal execution of jumps, movements, throws, pushes, and directional changes in both offensive and defensive phases of the game (*Karcher & Buchheit, 2014*; *Balsalobre-Fernández & Torres-Ronda, 2021*). When assessing muscular strength, it is essential to employ evaluations capable of discerning analytical improvements in the athlete's muscular strength generation, as well as its transfer and application to the game (*Abuajwa et al., 2022*; *Cavedon, Zancanaro & Milanese, 2018*; *Ferragut et al., 2018*; *Hammami et al., 2021*; *Starczewski, Borkowski & Zmijewski, 2020*; *Wagner et al., 2014*, *2017*; *Chirosa-Ríos et al., 2023*).

Functional electromechanical dynamometry (FEMD) is a device that is designed to train and assess variables such as speed and strength in sports gestures (*Rodriguez-Perea et al., 2021*) for different manifestations of muscular strength and types of contraction in free, single, or multi-joint movements of the whole body (*del-Cuerpo et al., 2023*). FEMD allows for dynamic (tonic, kinetic, elastic, inertial, and conic) or static (isometric and vibratory isometric) mode operation, enabling assessment and training through constant and variable resistance/velocity (*Jerez-Mayorga et al., 2020*; *Rodriguez-Perea et al., 2021*).

A recent study (*Morenas-Aguilar et al., 2023*) determined the relative and absolute reliability of a functional test with FEMD of three specific strength tests in handball players, showing an acceptable to high relative reliability in all exercises for average muscular strength and maximum muscular strength with an ICC > 0.83 and with a coefficient of variation (CV) <10%. *Baena-Raya et al. (2023)* reported high reliability in peak muscular strength (ICC = 0.94–0.95; CV = 2.22–2.51%) for the mid-thigh pull exercise with FEMD in isometric mode. Another study demonstrated high reliability (ICC = 0.95–0.98) and stable repeatability for the protocols used (CV < 10%), for muscular strength and velocity of movement of the concentric phase of five sit-to-stand measures, using three incremental loads controlled by a FEMD in healthy young adults (*Jerez-Mayorga et al., 2021*).

However, so far, the relative and absolute reliability of studies with FEMD has only been demonstrated in exercise protocols for inter-session comparison (test-retest), and no study has been found that has analyzed the relative and absolute reliability of intra-session sets for an exercise of isometric mean and peak muscular strength of the upper limbs in handball players with FEMD and that approximates blocking, pushing, and fixing actions. We sought an exercise widely used for strength development, such as the Bench Press. The idea was to adapt it to the sports context, which led us to the use of FEMD, providing the opportunity to perform the bilateral standing press (BSP). This exercise mirrors the motor gestures observed in handball gameplay, including blocking actions in the defensive phase, pushing, and fixation.

Therefore, the objectives of the study were to analyze the relative and absolute reliability of intra-session comparisons in a three-set protocol for maximal isometric strength (as a reflection of muscle strength) in a bilateral standing press (BSP) exercise among female handball players.

## MATERIALS AND METHODS

### Participants

Sixteen young international female handball players (19.65 ± 2.9 years, body mass 67.85 ± 6.59 kg, height 1.66 ± 0.07 m, body mass index (BMI) 23.44 ± 1.52 kg/m$^2$) with no experience in the use of functional electromechanical dynamometers (FEMD) voluntarily participated in the study. All had at least 5 years of competitive and strength training experience. None of the players had physical limitations, health problems or musculoskeletal injuries that could compromise the test. Prior to testing, participants were informed of the purpose and procedures of the research. All gave written consent to participate in the study. The study and intervention protocol adhered to the principles of the Declaration of Helsinki (*World Medical Association, 2013*) and was approved by the Institutional Review Board of the University of Granada (IRB approval: 3074/CEIH/2022).

### Experimental design

A repeated measures design was used to assess the relative and absolute reliability of a test of mean and peak isometric force in BSP exercise. All participants continued with their regular training programmed during the study but were asked to refrain from any strenuous physical activity for at least 24 h prior to the test. All assessments were conducted by the same assessor who had experience using the FEMD for more than 2 years. The study protocol was conducted at the same time of day (±1 h) and in similar environmental conditions (>21 °C and >60% humidity) for all participants.

### Materials

Isometric strength data were measured with a FEMD (Dynasystem, Model Research, Granada, Spain) which was adjusted for displacement, sensed load, and sampling rate as previously described by *Andrades-Ramírez et al. (2024)*.

## Testing procedures

Prior to data collection, participants attended two 60-min sessions to familiarize themselves with the assessment procedure and the use of the FEMD. In both sessions, the participants performed 10 sets of 5 s duration for the BSP exercise at an intensity of 50% of the perception of effort (Dynasystem Research, SYMOTECH, Spain, 2022). Instructions to participants were always the same and feedback was never given.

Height and body mass were measured at the first session with a wall-mounted stadiometer (Seca 202 Stadiometer; Seca Ltd., Hamburg, Germany) and a scale (Tanita Ironman BC-530TM, Japan). Prior to the sets, all participants performed a general warm-up consisting of 5 min of low-intensity jogging (heart rate per minute <130; measured with a Polar M400) and 5 min of joint mobility of the upper and lower limbs. The specific warm-up consisted of four sets of 5 s of submaximal isometric force with FEMD, coupled with a simple pulley and grip system.

## Isomteric force assesment (BSP)

Participants performed the exercise in a standing position with an anterior projection of the dominant foot of 20 to 30 cm, knees kept semi-flexed and hip extension of the non-dominant limb of 15° to 20°, which was controlled for all repetitions by visual feedback. Participants were instructed to avoid anterior trunk tilt (0° inclination from vertical). Bilateral upper limb anterior thrust was performed in shoulder abduction and 90° elbow flexion respectively and a pronated hand position was maintained (Fig. 1). During the session, three repetitions of maximum isometry of 5 s will be performed with a pause of 3 min between each repetition. Participants were instructed to exert as much force as possible for 5 s and hold the position for each repetition. The results were extracted from the device into a digitized spreadsheet for further analysis.

## Statistical analysis

Descriptive statistics were used to calculate means and standard deviations (SD) for the maximal muscle force assessment tests. The normal distribution of the data was analyzed using the Shapiro-Wilk test. Comparisons between series were carried out using one-way repeated measures ANOVA. The effect size ($\omega^2$) was considered as trivial (<0.01), small (0.01), medium (0.06) and 0.014) large. Paired-samples t-test and standardized mean differences (effect size (Cohen's d) for repeated samples) were used to compare mean and peak force between sets. The criteria for interpreting ES magnitude were as follows: null (<0.20), small (0.2–0.59), moderate (0.60–1.19), large (1.20–2.00) and very large (>2.00) (*Hopkins, 2017*). Relative reliability was assessed using the model intraclass correlation coefficient (ICC) and absolute reliability was assessed using the coefficient of variation (CV) and standard error of measurement (SEM). The following criteria were used to determine acceptable (CV < 10%, ICC < 0.80) and high (CV < 5%, ICC < 0.90) reliability (*Weir, 2005*). Bland-Altman plots were constructed to explore the concordance of the FEMD with respect to muscle force assessments and quantify systematic bias and 95% limits of agreement between series (*Boehringer & Whyte, 2019*). Heteroscedasticity of errors in the Bland-Altman plot was defined as a coefficient of determination ($R^2$) < 0.1

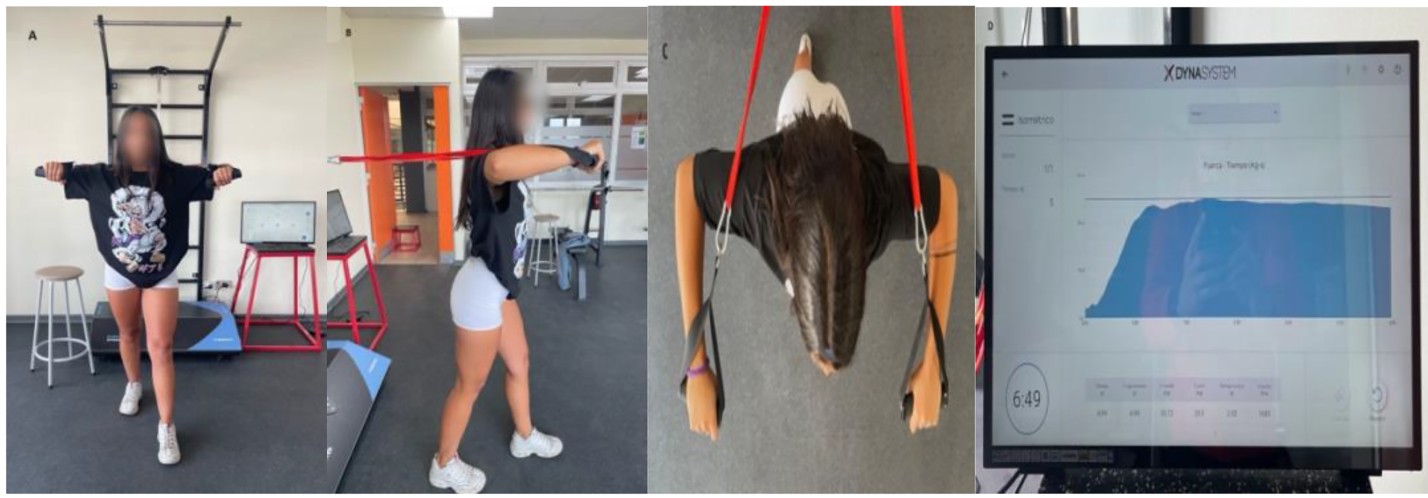

**Figure 1** **Bilateral standing press.** Standing bilateral press (BSP): (A) frontal plane view; (B) sagittal plane view; (C) horizontal plane view; (D) the record of isometric muscle strength levels during the 5 s FEMD tests is shown.

(*Atkinson & Nevill, 1998*). Finally, Pearson's correlation coefficient (Pearson's r) was used to quantify the correlation of all outcome variables between the three assessment sets. The criteria for interpreting the magnitude of $r$ were null (0.00–0.09), small (0.10–0.29), moderate (0.30–0.49), large (0.50–0.69), very large (0.70–0.89), near perfect (0.90–0.99), and perfect (1.00) (*Hopkins et al., 2009*). For all statistical calculations, a confidence interval of 95% was used in the analysis. Statistical significance was accepted at $p < 0.05$. All reliability assessments were conducted using a customized spreadsheet (*Hopkins, 2017*), while other statistical analyses were conducted using JASP software (version 0.16.4).

## RESULTS

No significant differences were reported in the comparison of means by groups with the repeated measures ANOVA statistic $\omega^2 = 0.001$ and no significant differences ($p > 0.730$), Furthermore, no differences were obtained were found in the inter-set analysis of assessments ($p > 0.05$ and ES < 0.20) for mean and peak isometric muscular strength and high reliability was obtained (ICC > 0.90; CV < 10%). The SEM ranged from 0.19 to 1.38 for mean muscle strength and from 0.29 to 2.79 in peak muscle strength as shown the Table 1.

Bland-Altman plots reveal a low systematic bias (−0.786–0.197) for mean isometric muscle strength and (−0.626–0.644) for peak muscle strength and $R^2 = 0.038$–0.169 in the set 1–set 2 evaluations, series 2–series 3 and series 1–series 3. As shown the Fig. 2.

A very large and significant association was found between set 1–set 2, set 2–3 and set 1–set 3 (0.807–0.895; $p < 0.001$) in mean strength and (0.867–0.889; $p < 0.001$) for peak muscular strength. As shown the Fig. 3.

## DISCUSSION

The present study was designed to analyze the relative and absolute reliability of intra-session comparisons in a three-set protocol for maximum isometric force in a
**Table 1 Relative and absolute reliability for intra-set comparison of mean and peak isometric muscle strength.**

| | Mean ± SD (kg) | | p-value | ES | ICC (95% CI) | SEM (%) (95% CI) | CV (95% CI) |
|---|---|---|---|---|---|---|---|
| **Set 1–Set 2** | | | | | | | |
| Mean force | 24.00 ± 6.69 | 23.80 ± 6.33 | 0.69 | 0.03 | 0.96 [0.89–0.99] | 1.38 [1.02–2.13] | 5.77 [4.26–8.93] |
| Peak force | 31.28 ± 10.61 | 30.63 ± 9.50 | 0.52 | 0.06 | 0.93 [0.82–0.98] | 2.79 [2.06–4.32] | 9.03 [6.67–13.97] |
| **Set 1–Set 3** | | | | | | | |
| Mean force | 24.00 ± 6.69 | 24.59 ± 6.15 | 0.31 | 0.09 | 0.95 [0.85–0.98] | 0.26 [0.19–0.40] | 6.59 [4.87–10.20] |
| Peak force | 31.28 ± 10.61 | 31.26 ± 8.92 | 0.98 | 0.01 | 0.93 [0.82–0.98] | 0.29 [0.22–0.45] | 8.82 [6.51–13.65] |
| **Set 2–Set 3** | | | | | | | |
| Mean force | 23.80 ± 6.53 | 24.59 ± 6.15 | 0.07 | 0.12 | 0.97 [0.92–0.99] | 0.19 [0.14–0.29] | 4.78 [3.53–7.41] |
| Peak force | 30.96 ± 9.68 | 31.26 ± 8.92 | 0.43 | 0.07 | 0.95 [0.87–0.98] | 2.21 [1.63–3.42] | 7.15 [5.28–11.28] |

Note:
SD, standard deviation; ES, Cohen d effect size; ICC, intraclass correlation coefficient; CV, coefficient of variation; SEM, standard error of measurement; 95% CI, 95% confidence interval.

bilateral standing press exercise (BSP) among female handball players. The results showed high relative (ICC > 0.90) and absolute (CV < 10%) reliability for mean and peak force, with no significant differences between sets ($p > 0.05$) reporting a null effect size (ES < 0.20). Which confirms that the BSP test is a reliable method for the evaluation of maximum isometric strength with FEMD.

In a previous study, similarities were reported with the results obtained in this study (*Sánchez-Sánchez et al., 2021*), which analyzed the reliability of a FEMD on measurements in an eccentric swing exercise in a linear isokinetic strength movement for the hamstrings. In soccer players, presenting a "high" reliability in the average force for the isokinetic speed of 0.4 $m \cdot s^{-1}$ (ICC = 0.94; CV = 2.80%). In the study (*Rodriguez-Perea et al., 2019*), which evaluated the maximum isometric strength of the trunk flexors with DEMF, results with high reliability in the relative measurements were observed (ICC = 0.71–0.83) in absolute concentric contraction and very high for eccentric contraction (ICC = 0.79–0.81) similar to this study. In *Martinez-Garcia et al. (2020)*, high to excellent reliability was observed in the internal rotator muscle strength of the shoulder with FEMD, reporting values of ICC = 0.85 and CV = 8.27% for the concentric phase and ICC = 0.81 and CV= 7.28% in the eccentric phase at a speed of 0.3 $m \cdot s^{-1}$, for the concentric phase values ICC = 0.93 and CV = 6.31% were obtained and in the eccentric phase ICC = 0.87 and CV = 6.87% at a speed 0.6 $m \cdot s^{-1}$, in addition an ICC = 0.90 and CV = 6.39% in the concentric phase, ICC = 0.98 and CV = 6.91% in the eccentric phase at a speed 0.3 $m \cdot s^{-1}$, for the concentric phase. The values obtained were ICC = 0.89 and CV = 6.26%, for the eccentric phase ICC = 0.92 and CV = 5.12%, which are very similar to those obtained in this research. The results of this research are comparable to the results of *Roth et al. (2017)*, who reported ICC = 0.91 and a CV = 7.3% when evaluating trunk strength in isometric flexion and obtained an ICC = 0.87–0.91 and a CV = 8.7% in trunk rotation using an isokinetic device (IsoMed-200).

In *Morenas-Aguilar et al. (2023)*, reliability measures similar to this study were observed, reporting the absolute and relative reliability of three specific strength tests in handball players with FEMD (standing lift–unilateral pullover–step forward), observing

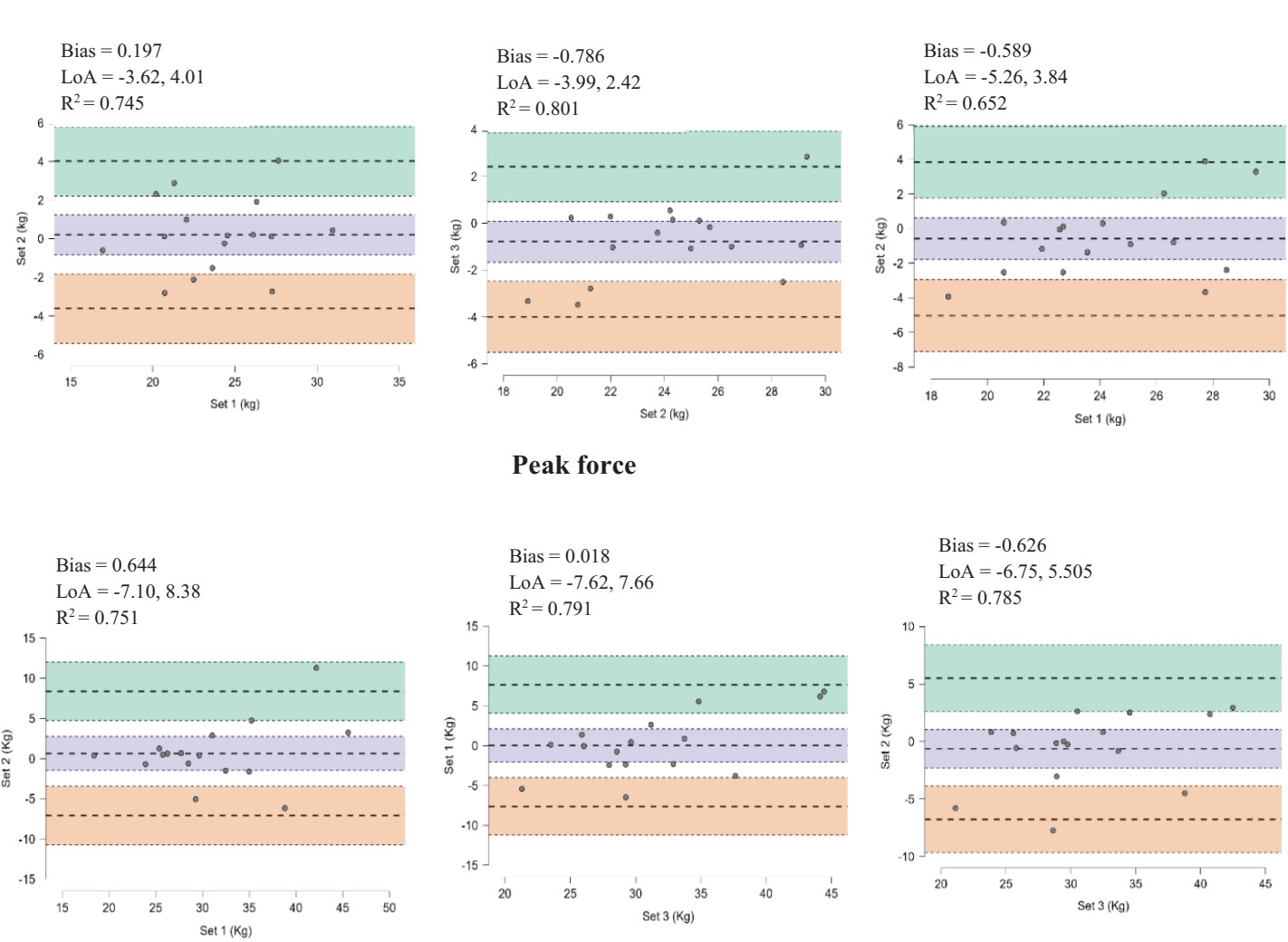

**Figure 2 Bland-Altman plot.** Bland-Altman plot reveal a low systematic bias (−0.786–0.197) for mean isometric muscle strength and (−0.626–0.644) for peak muscle strength in the set 1–set 2 evaluations, series 2–series 3 and series 1–series 3.

high reliability, or acceptable reliability, for mean and maximum strength of unilateral pullover, standing raise, and forward step (ICC range = 0.83–0.97; CV range = 3.90–11.57). Significant differences were found between the mean and maximum force (CV ratio > 1.16) and the right and left sides for the mean force in all exercises (CV ratio > 1.39) and the maximum force for the standing lift (CV ratio = 1.24).

These new functional electromechanical devices need an adequate familiarization process to ensure repeatability of measurements in the assessments of different manifestations of strength, especially with variable movements (*Sánchez-Sánchez et al., 2021*). In a study conducted with young handball players, throwing skills showed poor reliability when no assessment familiarization process was completed (*Koopmann et al., 2022*). In our research, two familiarization sessions were carried out, with which we ruled
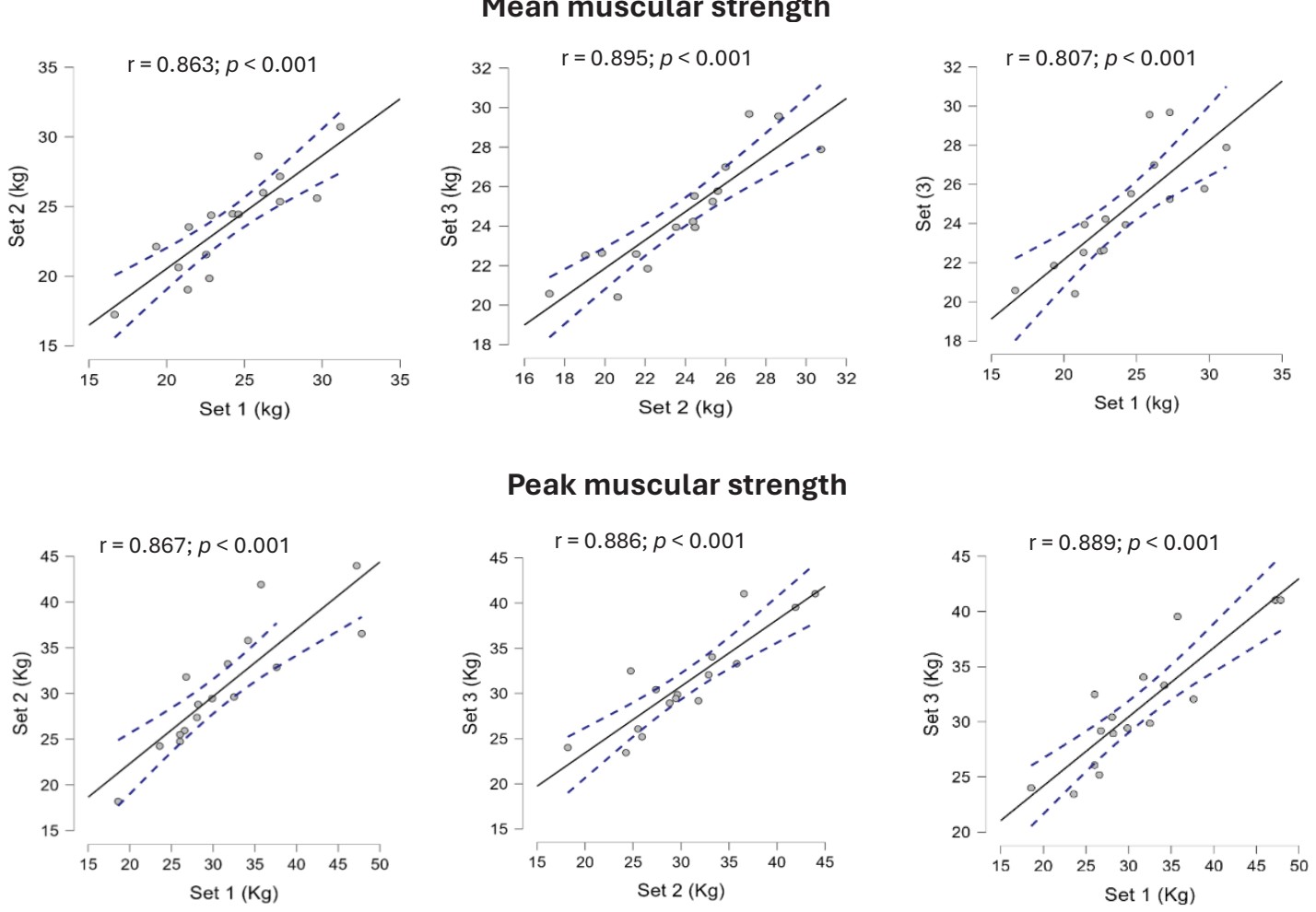

**Figure 3 Pearson correlation.** Pearson correlation (r) between intra-sets for isometric maximal strength of bilateral standing press.

out the possibility that the learning of the motor gesture has an influence on the generation of muscular force of the evaluated subjects.

The application of this BSP protocol and its approach to the sports gestures of blocks and fixations in the defensive phase of the handball players, allows a more precise evaluation when evaluating the muscular force and orienting the training processes of the upper body. Based on the obtained data, we can determine that the FEMD presents another reliability when it is intended to evaluate muscle strength in a specific and functional application of handball.

It should be considered that the experimental sample is international handball athletes and is not large enough to extrapolate the data obtained to the whole population or other sports. The demonstration of the reliability of the FEMD will allow coaches and trainers using this training and assessment technology to have a strength assessment exercise protocol.

## CONCLUSIONS

Based on the results of this study, it can be concluded that the FEMD presents a high relative and absolute inter-set reliability to determine the isometric strength for a BSP exercise in handball players. This can facilitate a more objective assessment of the specific function of the upper limbs in defensive actions of blocking, pushing, and fixing.

### Strengths and limitations of the study

There is sufficient evidence on inter-session reliability with the use of FEMD. However, there was no report analyzing the relative and absolute reliability in any of the FEMD device modes, this being the first study to analyze it. A second strength is that the exercise used is close to frequent sporting gestures in handball. One of the limitations is that our results are only reproducible for the isometric mode, further evidence is required on the relative and absolute intra-session reliability of the device modes.

### Funding

The authors received no funding for this work.

### Competing Interests

The authors declare that they have no competing interests.

### Author Contributions

- Claudio Cifuentes-Zapata conceived and designed the experiments, performed the experiments, analyzed the data, prepared figures and/or tables, and approved the final draft.
- Oscar Andrades-Ramírez performed the experiments, analyzed the data, prepared figures and/or tables, and approved the final draft.
- David Ulloa-Díaz conceived and designed the experiments, prepared figures and/or tables, and approved the final draft.
- Ángela Rodríguez-Perea analyzed the data, authored or reviewed drafts of the article, and approved the final draft.
- Álvaro Huerta Ojeda conceived and designed the experiments, authored or reviewed drafts of the article, and approved the final draft.
- Luis Javier Chirosa-Ríos analyzed the data, authored or reviewed drafts of the article, and approved the final draft.

### Human Ethics

The following information was supplied relating to ethical approvals (*i.e.*, approving body and any reference numbers):

Institutional Review Board of the University of Granada (IRB approval: 3074/CEIH/2022).

## Data Availability

The raw data are available in the Supplemental Files.

## Supplemental Information

Supplemental information for this article can be found online at http://dx.doi.org/10.7717/peerj.18196#supplemental-information.

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
