# Peer review of "Intra-session reliability of isometric muscle strength of the bilateral standing press in female handball players"

_PeerJ, doi:10.7717/peerj.18196_

## Round 0.1 · original submission · Minor Revisions

I congratule the authors on a well written manuscript. The manuscript is of high relevance but the reviewers have some concerns which I encourage the authors to address.

Reviewer 1 ·

Basic reporting

The aim of this study was to analyze the relative and absolute reliability of intra-session comparisons in a three-set protocol for maximum isometric force in a bilateral standing press exercise (BSP) among female handball players. The authors considered the two most common parameters, Fmax and Fmean, to describe the results of force recordings using a dynamometric system during the execution of the exercise.
From my perspective, the work is written and structured correctly. This work is interesting because it addresses an essential step for subsequent analysis of the effects of various factors on exercise, that is often overlooked. However, I allow myself to offer some suggestions that I believe would enhance the paper before Acceptance.

Abstract:
In broad terms, this section is accurate; I only suggest making two minor adjustments.
Firstly, specify the analyses conducted to determine the relative and absolute reliability. Secondly, the authors should clarify, before presenting the main results, that two force parameters were considered (maximum and average), given that the exercise was conducted under maximum isometric conditions, which could potentially cause confusion among readers.
Third, change CV to SEM in the results indicated as a reflection of absolute reliability.

Introduction:
The literature cited is relevant, I think minor adjustments are needed regarding the sequence in which references appear in some citations. It would also be necessary to review the correct usage of the term "force" or "strength." One term is more closely associated with force manifestation, while another refers to the force generation. The choice of these terms depends on the context in which it is used, force should be used when referring to the physical measurement itself and strength can be used if referring to muscular actions. Please review the second, third, and fourth paragraphs of the introduction and ensure consistency in terminology, as they also reference force generation and manifestations of muscular strength.
It would be beneficial to elaborate what was expressed in line 124 by indicating the significance of this information.

Results:
Line 212, the abbreviation for standard error of measurement was already indicated on line 193, you should use it here.
There is an error in Figure 2; it should say "peak" instead of "peck."

Discussion:
From what is expressed in the discussion, many previous studies have used CV to analyze reliability, which makes me understand why the authors have included it. However, It should be included in the discussion the values found for SEM.
Additionally, the results from the Bland-Altman plots and correlations conducted are not discussed.

Experimental design

The research question is clearly defined and, as I have indicated before, it is relevant.
Overall, the methods are clear, easy to understand, and replicable, with classic statistical procedures. Although regarding the latter, I would suggest instead of performing a pairwise t test to perform a repeated measures ANOVA with post-hoc test. Although paired t-tests are simpler to interpret, as they directly compare two conditions at a time, repeated measures ANOVA controls type I error more effectively than paired t-tests because it considers the correlation between repeated measurements. Additionally, it utilizes all available information from repeated measurements to estimate variability and detect differences between conditions. This also implies using another metric to estimate the effect size (Eta, partial eta or omega square).

Furthermore, the coefficient of variation (CV) is not a direct indicator of the absolute or relative reliability of a measure. The statistic commonly used to quantify absolute reliability is the SEM, which the authors also considered.

Validity of the findings

As indicated in the basic report, this work is interesting because it addresses an essential step for the subsequent analysis of the effects of various factors on exercise, which is often overlooked. Furthermore, the analysis is carried out using a device that has great potential for the quantitative analysis of the exercise.
The results obtained are consistent with the analyzes carried out by the authors, although as I already expressed, I suggest making changes in this aspect. Likewise, I suggest expanding the discussion considering the results of SEM, results from the Bland-Altman plots and correlations.

Additional comments

no comment'

·

Basic reporting

Good writing style and skill.

Reporting is good, although some additional details regarding statisticial analysis and other methods are required.

Some minor errors in the figures exist.

Experimental design

The main issue is why the authors did not analyse the familiarization sessions along with session 3 to determine inter-session reliability along with intra.

Validity of the findings

No issues.

Additional comments

General:
The study is well done, and the manuscript is clear, concise, and well-written. While the topic is not groundbreaking, the examined test is somewhat novel. Therefore, a simple reliability study is required for the literature in this area to move forward.
There are some important issues. These include not providing enough background as to why the test itself is valuable/worth studying, what the familiarizations consistent of, why the familiarizations weren’t analysed to determine inter-session reliability, and highlighting the limitations and what to do about them.
Please see section-specific comments below.

Title:
No issues, but it might read better as “Intra-session reliability of bilateral standing press isometric force in female handball players”. Same meaning, but slightly fewer words

Abstract:
Capitalize the first word following each subheading.
Overall the abstract is good. A few points to consider.
I recommend the authors slightly re-structure and/or include a brief sentence regarding why the BSP is potentially valuable for assessing (handball) athletes.
What is the specific effect size that the authors mention at the end of the ‘results’?

Introduction:
Nicely written and laid-out.
My main issue with the introduction is that the examined test itself (BSP) is only really mention in the very last paragraph, with very little information regarding why/how it is relevant and worth studying. This should be clear!
In paragraph 3, the authors first bring up ‘FEMD’. However, I am not exactly sure what this is. I know from reading the rest of the paragraph that it is not like an isokinetic dynamometer, but that is about it. Do FEMDs include strain-gauges/compression-gauges? If so, perhaps including that information in brackets or similar would be valuable and add needed context.
In line 105, the authors mention ‘kinetic’ in the bracket describing ‘dymanic’ operation. However, aren't ‘kinetics’ things like force and torque, whereas ‘kinematics’ include inertia, velocity etc.? I believe I am right, but I am not 100% sure and am open to an explanation from the authors.

Methods:
Two familiarization session are valuable, nice job!
Can the authors provide more information regarding the familiarization sessions? How many sets/reps? What intensities etc.
Were maximal efforts measured during the familiarization sessions? If so, it would be exceptionally valuable to include comparisons between the 2-3 sessions so that the paper also reports ‘inter’ session reliability! Indeed, this is likely more important than ‘inter’.
Line 190: I imagine that these are Cohen’s d effect sizes. If so, please be clear about this.
Good/thorough statistical analysis.

Results:
No issues here, although see below regarding the figures.

Discussion:
Good discussion.
My only really issue is the lack of a paragraph focusing on the limitations and how to address them. For example, determining intra-session reliability is important, but inter-session reliability is MORE important.

Figures/tables:
While figure one does a good job of illustrating the testing position, I highly recommend that an additional figure, or panel, be added that shows the device itself up close.
The table and other figures are good!
Figure 2 says ‘Peck force’ instead of ‘Peak force’ above the bottom 3 BA-plots. Same with figure 3

Reviewer 3 ·

Basic reporting

Thank you for the opportunity to review the study.

My observations are as follows:

1- Figure 1 B – it should be review it, in the bench press exercise (standing or vertical) the hand is pronated. The description “neutral hand position” are presented in abstract and methods.

2 - Bland-Altiman plots Y-axis is the difference of the measures and X-axis is the average (mean) of the measures. Therefore, the labels axis has to change.

3 - Raw data – mean force is not available;

4 - Please share the Hopkins's customized spreadsheet with your analyses;

5 - Describe which ICC type was used, also CV and SEM formulas were used.

Experimental design

No comment.

Validity of the findings

Please, provide the raw data of mean force and the reliability analyses spreadsheet.

Include the strengths and limitations of the study.

Additional comments

Overall, the study is well-written. However, some areas require improvement for optimal reader comprehension:

The idea presented in lines 266 to 270 needs a concluding statement.

The paragraph (line 272 to 275) lacks cohesion within the discussion and requires tighter integration of ideas.

---

## Round 0.2 · Minor Revisions

The authors did well on implementing all major comments. There are now still some minor comments which I encourage the authors to address carefully before a accept decision will be made.

Reviewer 1 ·

Basic reporting

In its current version, aside from the minor comments included in item 4, I find the article to be engaging, clear, and well-structured.

Experimental design

In its current version, the manuscript is accurate and clear regarding its experimental design

Validity of the findings

I have no comments or concerns regarding this section

Additional comments

Regarding the adjustments made by the authors in response to my previous comments, I would like to highlight the following aspects:

Firstly, concerning the use of the terms "strength" and "force," my suggestion was not to standardise the terminology but rather to use the appropriate term in each context. In this regard, I believe that in line 71, where a variable is mentioned, the term "force" should be used. Additionally, in the same sentence, I recommend removing the term "kinetic" from the variables. This term may cause confusion for some readers, as the set of variables includes both force and speed, and the sentence remains clear without it.

Regarding the terminology, in the objective section, the precise phrasing would be:
… the objectives of the study were to analyse the relative and absolute reliability of intra-session comparisons in a three-set protocol for peak isometric force (as a reflection of muscular strength) in a bilateral standing press exercise (BSP) among female handball players.

I leave this decision to the authors and the editor, as despite the existence of relevant articles (e.g., Winter EM, Abt G, Brookes FB, Challis JH, Fowler NE, Knudson DV, Knuttgen HG, Kraemer WJ, Lane AM, van Mechelen W, Morton RH, Newton RU, Williams C, Yeadon MR. Misuse of "Power" and Other Mechanical Terms in Sport and Exercise Science Research. J Strength Cond Res. 2016 Jan;30(1):292-300. doi: 10.1519/JSC.0000000000001101. PMID: 26529527), these issues have not yet been fully addressed in the field of sports sciences.

Finally, on line 85, there appears to be an issue with the order of the terms; it should correctly read "muscular strength."

·

Basic reporting

No issues

Experimental design

No major issues.

While I disagree with the authors not analyzing the familiarization session to determine learning effects, they do answer my issue and provide good rationale.

I do not believe that publication should be denied on this point.

Validity of the findings

No issues.

Reviewer 3 ·

Basic reporting

no comment.

Experimental design

no comment.

Validity of the findings

no comment.

Additional comments

The BA plots in the review PDF differ from the tracked changes file (96649.docx).

---

## Round 0.3 · accepted · Accept

Thank you. All comments were satisfactorily addressed. Congratulations